# Electrochemical carbon–carbon coupling with enhanced activity and racemate stereoselectivity by microenvironment regulation

Kejian Kong[1], An-Zhen Li[1], Ye Wang[1], Qiujin Shi[1], Jing Li[1], Kaiyue Ji[1] & Haohong Duan [1,2,3] ✉

Enzymes are characteristic of catalytic efficiency and specificity by maneuvering multiple components in concert at a confined nanoscale space. However, achieving such a configuration in artificial catalysts remains challenging. Herein, we report a microenvironment regulation strategy by modifying carbon paper with hexadecyltrimethylammonium cations, delivering electrochemical carbon–carbon coupling of benzaldehyde with enhanced activity and racemate stereoselectivity. The modified electrode–electrolyte interface creates an optimal microenvironment for electrocatalysis—it engenders dipolar interaction with the reaction intermediate, giving a 2.2-fold higher reaction rate (from 0.13 to 0.28 mmol h$^{-1}$ cm$^{-2}$); Moreover, it repels interfacial water and modulates the conformational specificity of reaction intermediate by facilitating intermolecular hydrogen bonding, affording 2.5-fold higher diastereomeric ratio of racemate to mesomer (from 0.73 to 1.82). We expect that the microenvironment regulation strategy will lead to the advanced design of electrode–electrolyte interface for enhanced activity and (stereo) selectivity that mimics enzymes.

Enzymes achieve atom-precise transformations of small molecules (e.g., $CO_2$ (ref. 1), $N_2$ (ref. 2)) or macromolecules (e.g., protein[3], cellulose[4]) by creating a confined nanoscale space, in which multiple components acting in concert to recognize and catalyze the reactants, enabling high efficiency and catalytic specificity toward specific products. Inspired by the nature design, synthetic catalysts have been recently developed with sophisticated structure to employ non-covalent interactions with the reactants in a manner similar to an enzyme-binding pocket[5,6]. Owning to the development of renewable electricity and operation under ambient conditions[7], it is highly attractive to develop enzyme-like electrocatalysts. However, creating such a configuration for an electrocatalyst requires the confinement of

reactants and reaction intermediates at electrode–electrolyte interface, which is conventionally an open-space environment. Moreover, the dynamic stability of the configuration under external potential and the influence of solvent molecules (e.g., water) on particular interaction with the reactants or reaction intermediates should also be considered, which increases the difficulty of achieving such configuration at electrode–electrolyte interface.

Recently, significant progress in developing enzyme-like electrocatalysts has been made for electrocatalytic $CO_2$ reduction. For example, Yang and co-workers reported that the combination of silver (Ag) nanoparticle and a detached layer of ligands created an optimal interlayer for intercalating desolvated cations, which in turn enhanced

[1]Department of Chemistry, Tsinghua University, Beijing, China. [2]Haihe Laboratory of Sustainable Chemical Transformations, Tianjin, China. [3]Engineering Research Center of Advanced Rare Earth Materials, (Ministry of Education), Department of Chemistry, Tsinghua University, Beijing, China. ✉e-mail: hhduan@mail.tsinghua.edu.cn

activity and CO selectivity[8]. Recently, Li and co-workers demonstrated Ag in conjunction with quaternary ammonium cationic surfactants afforded an interfacial microenvironment, showing enhanced CO generation by repelling isolated water and enriching $CO_2$ molecules[9]. These successful examples show the potential of microenvironmental regulation to enhance catalytic activity. In addition, this strategy exhibits a wide range of applications[10,11], including electrocatalytic reductions (e.g., hydrogen evolution reaction (HER)[12], oxygen reduction reaction (ORR)[13] and carbon dioxide reduction reaction ($CO_2$RR)[14]) and oxidations (e.g., oxygen evolution reaction (OER)[15] and glycerol electrooxidation (GOR)[16]). Moreover, besides activity improvement, stereoselectivity control is also important for catalytic transformation, particularly in organic synthesis to afford valuable chemicals. However, stereoselectivity regulation of electrocatalytic aromatic aldehyde C–C coupling was rarely studied[17], with a lack of in-depth understanding of the structure-activity correlation. Meanwhile, microenvironment regulation has never been explored for stereoselectivity control in electrocatalysis, the success of which relies on the capability of the microenvironment to modulate the conformational specificity of the reaction intermediates.

Carbon–carbon (C–C) bond formation serves as a versatile and important synthetic strategy in organic chemistry[18]. Among various C–C bond formation reactions, pinacol C–C coupling of carbonyl groups shows attractiveness in producing chemicals and fuels that find widespread applications in medicines, pharmaceuticals and transportations[19]. Moreover, biomass-derived aldehydes, such as furfural, 5-hydroxymethylfurfural and benzaldehyde, can be utilized as the carbonyl feedstocks for C–C coupling, enabling renewable carbon sources for pinacol synthesis. For instance, pinacol C–C coupling of HMF with subsequent ring opening and hydrogenation reactions in tandem afford $C_{10}$–$C_{12}$ biofuels[20]. In recent years, the electrochemical approach has stimulated increasing attention for its merits of using renewable electricity to drive the reaction and using water as the proton source[21]. However, it is difficult to enhance the adsorption of reactants with promoted activity by electrode design, probably because pinacol C–C coupling follows an outer-sphere electron transfer process wherein the reactant is not directly adsorbed over the electrode surface[22].

Moreover, it should be noted that the produced pinacol (e.g., hydrobenzoin from benzaldehyde, hydrofuroin from furfural)

contains two chiral centers, thus structurally giving rise to three stereoscopic configurations, including RS, RR and SS[23] (Supplementary Fig. 1). Nevertheless, stereoselectivity control in electrochemical pinacol C–C coupling have been largely overlooked in previous studies. Considering the importance of chiral hydrobenzoin in the synthesis of value-added chemical intermediates[24], it would be attractive to achieve stereoselectivity control in electrochemical C–C coupling.

Herein, we report a microenvironment regulation strategy for enhanced activity and racemate stereoselectivity in electrochemical pinacol C–C coupling reaction. We first show that moderate reaction rate and mesomer stereoselectivity are given over bare carbon paper (CP) as the electrode (Fig. 1a). By modifying CP with cetyl trimethyl ammonium bromide (CTAB), we observed that the reaction rate increases by 2.2-fold (from 0.13 to 0.28 mmol $h^{-1}$ $cm^{-2}$), and the diastereomeric ratio (dr) of racemate to mesomer increases by 2.5-fold (from 0.73 to 1.82) at −1.4 V versus Ag/AgCl (Fig. 1b). We unveil that CTAB molecules undergo a potential-dependent conformational transformation, dynamically forming an ordered arrangement under reaction conditions, with its positively charged head toward the electrode interface and the alkyl chain toward the electrolyte. We demonstrate this configuration creates an optimal microenvironment to confine and modulate the configuration of the reaction intermediate at electrode–electrolyte interface (Fig. 1c). It engenders strengthened dipolar interactions with ketyl radical that promotes activity. Moreover, the hydrophobic interface repels water and facilitates intermolecular hydrogen bonding between the hydroxyl groups of ketyl radicals, hence modulating conformational specificity with enhanced racemate stereoselectivity.

## Results
### Catalytic performance of CTAB-modified carbon paper
For electrochemical pinacol C–C coupling, we selected CP as the electrode because of its low cost, high conductivity and large surface area. The characterization of CP was shown in Supplementary Fig. 2. We sought to modify CP electrode–electrolyte interface with CTAB, based on two considerations: (1) Owning to the amphiphilic property stemming from the positively charged head group (trimethyl ammonium) and a hydrophobic tail (hexadecyl group), CTAB shows the ability to modulate the microenvironment through interacting with

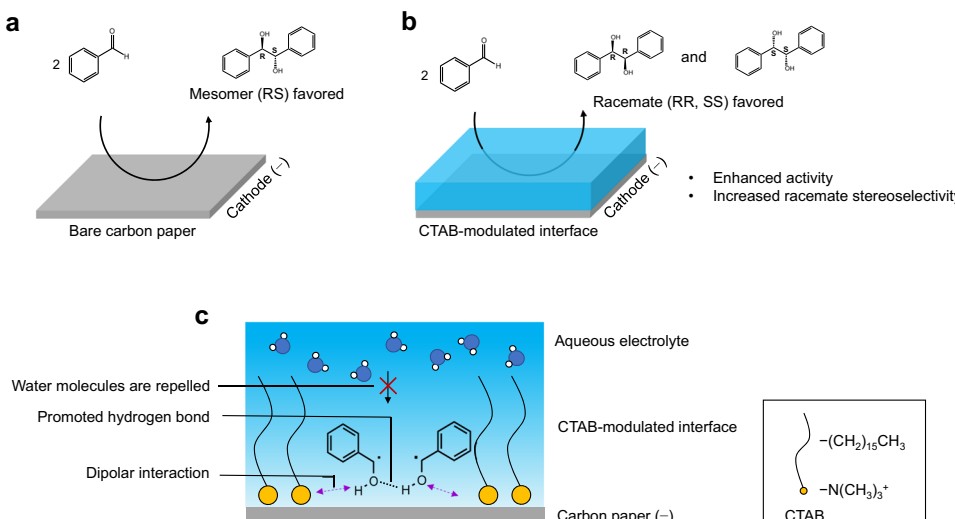

**Fig. 1 | Microenvironment regulation for electrochemical pinacol coupling of benzaldehyde.** Production distribution over **a** carbon paper and **b** CTAB-modified carbon paper. **c** Microenvironment at electrode–electrolyte interface of CTAB-modified carbon paper: confinement of ketyl radical via dipolar interaction with enhanced activity; modulation of conformational specificity of ketyl radical via repelling interfacial water and promoted hydrogen bond between ketyl radicals with enhanced racemate stereoselectivity.

reaction intermediates using its head group[25] or through enriching hydrophobic reactants using its hydrophobic tail[26]. We surmise that modification of CP with CTAB might modulate the C−C coupling reaction by interacting with ketyl radicals. (2) Because electrochemical C−C coupling takes place at the cathode, we anticipate that the positively charged head group of CTAB is conducive to its stabilization at the negatively charged interface under electrochemical conditions.

With these considerations in mind, we commenced our study by introducing 1 mM CTAB into 0.5 M $Na_2SO_4$ electrolyte containing 25 mM benzaldehyde using CP as the working electrode (the electrode is denoted as CP-CTAB). Bulk electrolysis experiments at −1.4 V were performed to evaluate the reaction rate and stereoselectivity by HPLC (Supplementary Figs. 3–6). After CTAB was introduced, the Faradaic efficiency (FE) of hydrobenzoin slightly increases (from 58.7 to 64.2%, Fig. 2b). Notably, the reaction rate of hydrobenzoin shows 2.2-fold improvement compared to that of CP (from 0.13 to 0.28 mmol $h^{-1}$ $cm^{-2}$, Fig. 2a). Moreover, dr of racemate/mesomer delivers 2.5-fold increase (from 0.73 to 1.82, Fig. 2b). To confirm the promoting effect, we investigated the catalytic performance at different potentials and CTAB concentrations. The reaction rate increases notably at a more negative potential (Fig. 2c), suggesting that electrochemical C−C coupling is facilitated at stronger reduction conditions. Compared with electrolysis results over bare CP electrode, introducing CTAB shows 4-fold increase of reaction rate at −1.3 V (from 0.04 to 0.16 mmol $h^{-1}$ $cm^{-2}$, Supplementary Fig. 7). Meanwhile, dr value of racemate/mesomer increases initially and then decreases at more negative potential (Fig. 2d). Regarding the effect of CTAB concentration, both the reaction rate (Fig. 2e) and stereoselectivity of racemates (Fig. 2f) increase steadily at higher CTAB concentrations, suggesting that CTAB indeed plays an important role in promoting activity and stereoselectivity of racemate. To eliminate the effect of bromide ion

(Br⁻) in CTAB, NaBr was used for electrolysis, and the reaction rate slightly decreased while the racemate slightly increased compared with those over bare CP (Supplementary Fig. 8), indicating that Br⁻ may not be the main reason for the enhanced activity and racemate stereoselectivity over CP-CTAB. To understand the promoting effect of CTAB, we selected a CTAB concentration of 1 mM that affords superior catalytic performance for the following studies.

## Potential-dependent conformational transformation of CTAB

The adsorption of CTAB over CP surface was demonstrated by linear sweep voltammetry (LSV, Supplementary Fig. 9), X-ray photoelectron spectroscopy (XPS, Supplementary Figs. 10 and 11), Fourier transform infrared spectroscopy (FT-IR, Supplementary Fig. 12), and X-ray diffraction spectra (XRD, Supplementary Fig. 13). Differential capacitance ($C_{diff}$) curve further indicates that CTAB molecules exhibit specific adsorption over CP via its positively charged head group (Supplementary Fig. 14) under electrochemical reaction conditions (Supplementary Fig. 15), with the formation of inner Helmholtz plane (IHP)[27], in agreement with the specific adsorption of Cs⁺, Tl⁺ ions (see Supplementary Note 1 for more discussion). We anticipated that the hydrophobic chain of CTAB would face toward the electrolyte side, thus creating a hydrophobic environment at the interface. To investigate whether the interface is hydrophobic, contact angle (CA) experiments were conducted (Supplementary Fig. 16). For the CP electrode without CTAB, it shows a contact angle of 113°, an indication of a hydrophobic interface. For the CP-CTAB electrode, however, it shows a contact angle close to 0°, suggesting that the interface becomes completely infiltrative when CTAB is adsorbed on CP. This finding seems contradictory to our expectation that the interface was hydrophobic after CTAB modification. We rationalize that since no external potential was applied in the CA experiments, the positively charged head group of

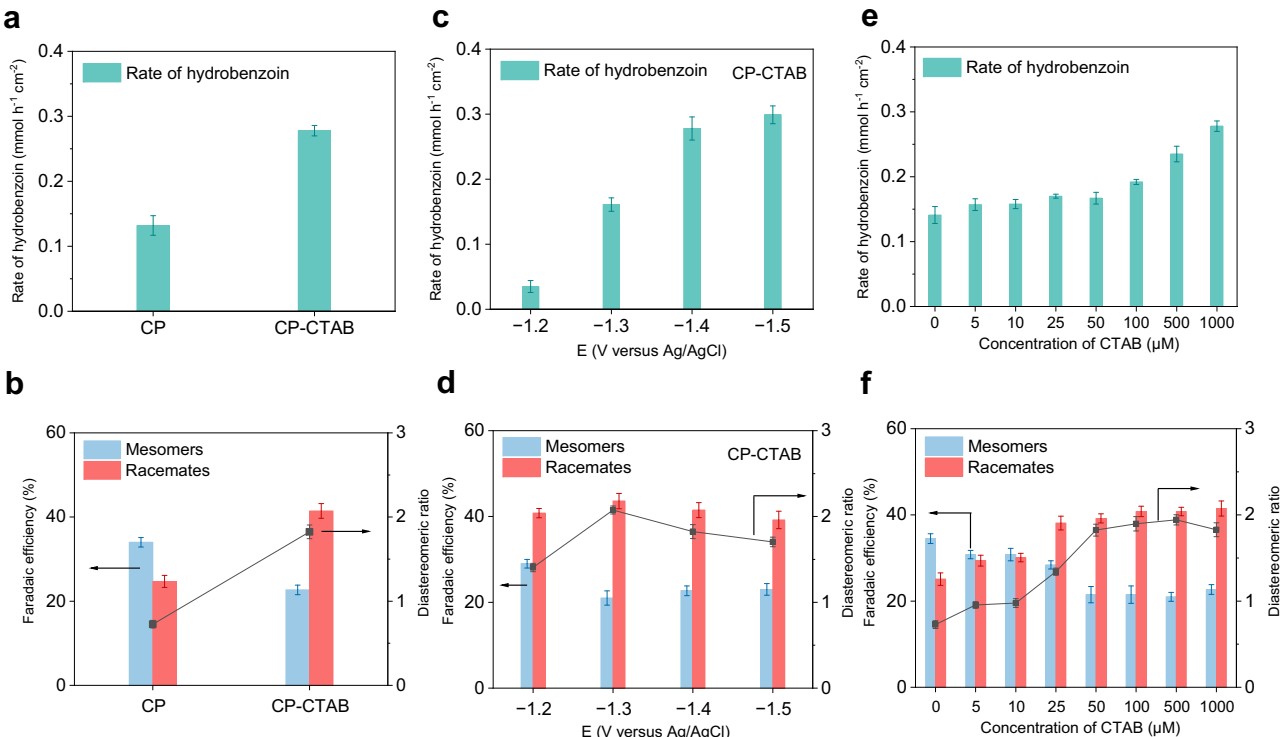

**Fig. 2 | Catalytic performance of CP and CP-CTAB in electrochemical pinacol coupling. a** Reaction rate and **b** stereoselectivity of hydrobenzoin over CP and CP-CTAB electrodes at −1.4 V versus Ag/AgCl with 1 mM CTAB in 0.5 M $Na_2SO_4$ electrolyte. **c** Reaction rate and **d** stereoselectivity of hydrobenzoin over CP-CTAB electrode at different potentials with 1 mM CTAB in 0.5 M $Na_2SO_4$ electrolyte. **e** Reaction rate and **f** stereoselectivity of hydrobenzoin over CP-CTAB electrode at −1.4 V versus Ag/AgCl with different concentrations of CTAB in 0.5 M $Na_2SO_4$ electrolyte. Error bars correspond to the standard deviation of three independent measurements.

CTAB may face the electrolyte (thus tail faces to CP interface), making the interface extremely hydrophilic, as observed in a previous report[28].

Based on these results, we speculate that CTAB might undergo conformational transformation at the CP surface depending on external potential, with its hydrophobic tail shifting from facing the electrode (without external potential) to facing the electrolyte (with external potential). To demonstrate this possibility, in situ attenuated total reflection surface-enhanced infrared absorption spectroscopy (ATR-SEIRAS) experiments were performed (Fig. 3a). At positive potentials, the band centered at 2951 cm$^{-1}$ appears, which is assigned to the symmetric C–H stretching vibrations of $CH_3$ species in the head group of CTAB. The asymmetric and symmetric C–H stretching vibrations of $CH_2$ species adjacent to the tail group (2916 and 2842 cm$^{-1}$, respectively) and head group (2921 and 2848 cm$^{-1}$, respectively)[29] of CTAB were also observed. When a negative potential was applied, the intensity of bands at 2916 and 2842 cm$^{-1}$ gradually decreased and those at 2921 and 2848 cm$^{-1}$ increased, suggesting that the head group of CTAB started to face the electrode when more negative potential was applied. To quantitatively analyze the variation of IR bands, Gaussian fitting was employed (Supplementary Fig. 17), and the results show that the population of the peak at 2916 cm$^{-1}$ (assigned to tail group) gradually decreases and the peak at 2921 cm$^{-1}$ (assigned to head group) increases (Fig. 3b), clearly demonstrating the potential-dependent conformational transformation of CTAB. This conclusion was further demonstrated by the fitting data of electrochemical impedance spectroscopy (EIS) experiments (Fig. 3c and Supplementary Fig. 18). A transition point was observed at approximately –1.0 V. When the potential is more negative than –1.0 V, the charge transfer resistance ($R_{ct}$) over CP-CTAB is lower than that over CP. Bode plot analysis affords a similar variation (Supplementary Fig. 19). These results can be rationalized by facilitated adsorption of CTAB at the CP interface with its positively charged head group because of electrostatic interaction at more negative potential. Therefore, charge was accumulated at the electrode–electrolyte interface, and electron transfer was accelerated, which consequently gives rise to a lower $R_{ct}$ value at more negative potential. The transition point at –1.0 V versus Ag/AgCl is consistent with in situ ATR-SEIRAS result (Fig. 3a), in which the signals of $CH_2$ adjacent to the tail group in CTAB almost disappeared at potential more negative than –1.0 V versus Ag/AgCl.

Based on the above evidence, we propose that the tail group of CTAB inclines to face to electrolyte while the head group faces the electrode at more negative potentials (Fig. 3d), leading to an accelerated electron transfer rate. Furthermore, we demonstrated that CTAB molecules are arranged orderly over the CP interface under electrochemical conditions because of the hydrophobic interaction between the hexadecyl groups in CTAB molecules (Supplementary Note 2, Supplementary Figs. 20–22). Under electrochemical conditions, the applied potentials (in the range of –1.2 to –1.5 V versus Ag/AgCl) are more negative than the transition point (–1.0 V versus Ag/AgCl), thus the induced higher electron transfer rate would be conducive to electrochemical pinacol C–C coupling. A more mechanistic study of the promoted reaction rate will be discussed later.

## Understanding of the promoted activity by CTAB adsorption

We then devoted to understanding the mechanism of promoted reaction rate in electrochemical benzaldehyde C–C coupling by CTAB adsorption over CP. Bear in mind that a factor that determines the rate-determining step (RDS) of a catalytic reaction would greatly affect the overall activity, thus identification of the RDS of electrochemical benzaldehyde C–C coupling is the prerequisite. The reaction is dominantly controlled by kinetics rather than diffusion, as evidenced by normal pulse voltammetry (NPV) experiments (Supplementary Fig. 23). We then demonstrated that the RDS involves transfer of the first electron to afford ketyl radical, following a proton-coupled electron transfer (PCET) process, as evidenced by Tafel experiment (Supplementary Fig. 24), electron paramagnetic resonance (EPR) and kinetic isotope effects (KIE) experiments (Supplementary Fig. 25).

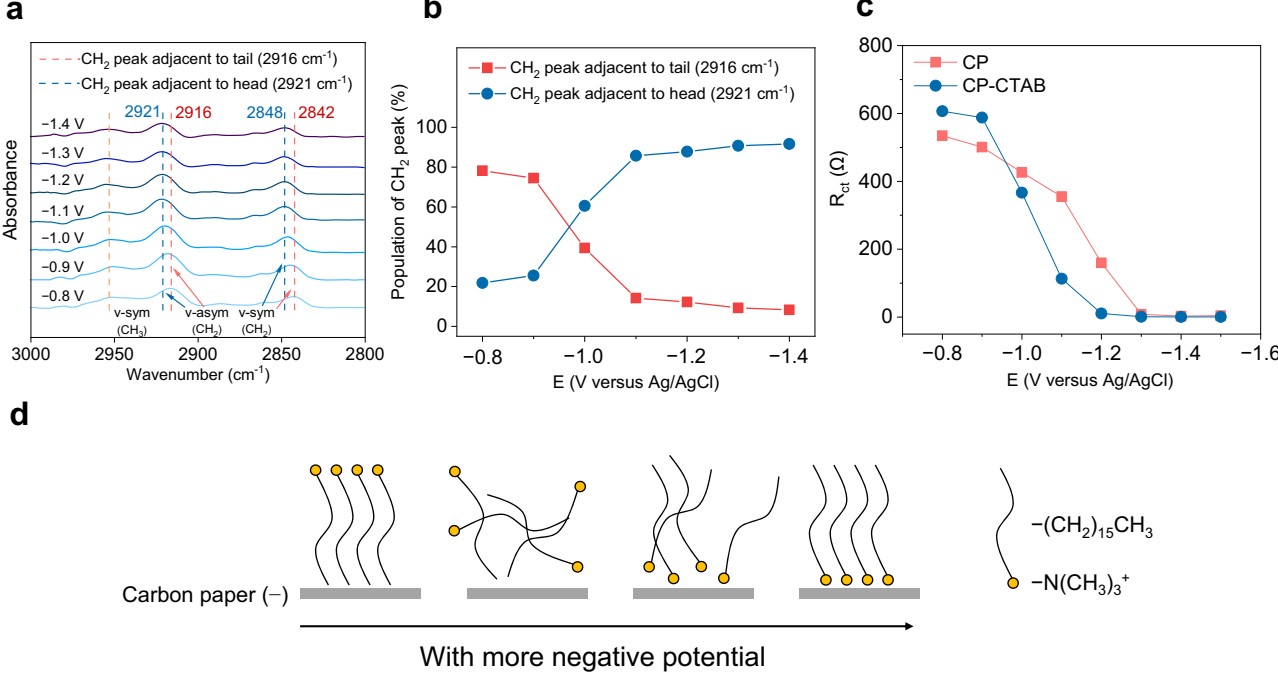

**Fig. 3 | Adsorption and conformation of CTAB at electrode–electrolyte interface. a** In situ ATR-SEIRAS spectra at different potentials over CP-CTAB electrode in the range of 3000–2800 cm$^{-1}$. **b** In situ ATR-SEIRAS spectra, showing potential-dependent population of $CH_2$ peaks in CTAB. **c** Resistance of charge transfer over CP and CP-CTAB electrodes at different potentials. **d** Schematic illustration of potential-dependent conformational transformation of CTAB molecules at electrode–electrolyte interface.

Therefore, we postulate that CTAB may play a significant role in modulating ketyl radical, thereby facilitating the reaction rate (see Supplementary Note 3 for more discussion).

In previous reports, different mechanisms were proposed for facilitated electrochemical catalytic activity by interface modification, including 1# buffer effect on local pH (ref. [30]), 2# modulation of local electric field (ref. [31]), or 3# interaction between CTAB and the reaction intermediates (ref. [32]). To understand how CTAB molecules modulate ketyl radical thus promoting the reaction rate in benzaldehyde C–C coupling in our work, we considered the above three possibilities.

Regarding possibility 1#, compared with Zhang's work[21], the difference in reaction rate between reactions occurring in neutral and alkaline mediums was negligible, reaching approximately 0.1 mmol h$^{-1}$ cm$^{-2}$ at a similar potential versus standard hydrogen electrode (SHE, Supplementary Fig. 26). Therefore, the effect of local pH variation by CTAB can be ruled out. In addition, considering H$_2$O is the proton source in the neutral medium, H$^+$ may not participate in the PCET process, thus local pH effect can be excluded.

Regarding possibility 2#, for quaternary ammonium cations (such as CTAB), the local electric field effect can be assigned to Frumkin correction with $\psi_1$ potential in the diffuse layer[33]. The specific adsorption of cations at electrode–electrolyte interface may result in an overload of anions in the diffuse layer, with the generation of reverse additional electric field, which in turn strengthens the electric field in IHP (due to constant applied potential) and enhances the reaction rate. In general, the size of the head group has a critical impact on the local electric field, and the intensity of the local electric field is inversely correlated with the size of the head group[34]. To study the local electric field effect, CTAB, hexadecyl dimethyl benzyl ammonium bromide (HDBAB) and ethyl hexadecyl dimethyl ammonium bromide (EHDAB) were selected to modify CP (Fig. 4a) because they have the same net-charge and chain length but different size of cation head groups (CTAB < EHDAB < HDBAB). We anticipate that different charge densities of head groups would be given with the sequence of HDBAB < EHDAB < CTAB, which in turn induces the same trend of local electric field intensity. As shown in Fig. 4b, introducing these quaternary ammoniums improves the reaction rate to a different extent, with the sequence of HDBAB < EHDAB < CTAB, which seems to support possibility 2# that the activity is promoted by a stronger local electric field. To rule out the influence of the long tails in these molecules, four spherical quaternary ammoniums with different head sizes but without long chains were employed (Fig. 4c), including tetramethyl ammonium bromide (TMAB), tetraethyl ammonium bromide (TEAB), tetra-n-propyl ammonium bromide (TPAB) and tetra-n-butyl ammonium bromide (TBAB). They display different local electric field intensities (TBAB < TEAB < TPAB < TMAB). However, the reaction rate is positively correlated with the size of the cation head group (Fig. 4d), suggesting that the local electric field might not be the key to the promoted activity.

Regarding possibility 3#, we surmise that the interaction between CTAB and the reaction intermediate (ketyl radical) may regulate the kinetics, thus facilitating the reaction rate. To investigate if the ordered arrangement of CTAB molecules is conducive to this assumption, we modified CP with quaternary ammonium cations containing the same head size but with different chain lengths to attain different arrangement order. Tetramethyl ammonium bromide (TMAB), butyl trimethyl ammonium bromide (BTAB), octyl trimethyl ammonium bromide (OTAB), dodecyl trimethyl ammonium bromide (DTAB) and CTAB were tested (Fig. 4e; carbon number of 1, 4, 8, 12, 16, respectively), and the catalytic results show that longer chain facilitates reaction rate (Fig. 4f). Since quaternary ammonium cations with longer chain show stronger hydrophobic interaction and display higher order of arrangement, the ordered arrangement of CTAB presumably facilitates the interaction with ketyl radical and promotes reaction rate. This rationale was further corroborated by the observation of quaternary

ammonium cations with smaller head group showing higher reaction rate (HDBAB < EHDAB < CTAB, Fig. 4b). This is because cations with smaller head group have less steric hindrance, giving rise to a higher order of arrangement.

## Interaction nature between CTAB and reaction intermediate

To understand the underlying mechanism of ordered arrangement of CTAB for enhanced interaction with ketyl radical, we sought to identify the interaction nature. We first demonstrated that hydrophobic interaction between the CP and ketyl radical is not a decisive factor for activity enhancement (Supplementary Fig. 27). We then assessed if dipolar interaction exists between the adsorbed CTAB and ketyl radical. According to previous reports on CO$_2$ electroreduction, ion-dipole interaction between CO$_2$ and alkali metal cations was demonstrated to enhance activity[35,36]. It was considered due to a short-range local interaction composed of both electrostatic and covalent interactions. If a similar dipole interaction between CTAB and ketyl radical exists that stabilizes the reaction intermediate, the PCET process in benzaldehyde C–C coupling can be modified according to Eq. (2).

$$PhCHO + e^- + H_2O + CTA^+ \rightarrow \cdot PhCHOH \cdots CTA^+ + OH^- \quad (1)$$

Based on this assumption, the reaction rate would display a linear correlation with the concentration of CTAB concentration at electrode–electrolyte interface according to the law of mass action[37]. Meanwhile, Gouy-Chapman model predicts that the concentration of charged species at electrode–electrolyte interface is proportional to the square root of the corresponding concentration in the bulk, according to Eq. (3).

$$[CTA^+]_{interface} = k'[CTA^+]_{bulk}^{\frac{1}{2}} \quad (2)$$

Based on the above two points, if there is dipolar interaction between CTAB and ketyl radical, the reaction rate would be linearly correlated with the square root of concentration of CTAB in the bulk[38], as deduced by Eq. (4).

$$r = k[Benz][CTA^+]_{interface} = k''[CTA^+]_{bulk}^{\frac{1}{2}} \quad (3)$$

where $k'' = kk'[Benz]$, and [Benz], [CTA$^+$]$_{interface}$ and [CTA$^+$]$_{bulk}$ refer to the concentration of benzaldehyde, CTAB at the interface and CTAB in the bulk, respectively. To our expectation, a linear correlation between reaction rate and the square root of [CTA$^+$]$_{bulk}$ was obtained (Fig. 4g), indicating that the dipolar interaction between CTAB and ketyl radical plays a decisive role in activity promotion (Fig. 4h). The dipolar interaction further explains the greatly reduced charge transfer resistance ($R_{ct}$) over CP-CTAB compared with that over bare CP (Fig. 3c), because the electron-transfer ability at electrode–electrolyte interface can be significantly accelerated via dipolar interaction between CTAB and CP.

Based on the above understanding, we posit that the ordered arrangement of CTAB molecules increases the coverage of the positively charged head group over the CP interface, stabilizing the ketyl radical via strengthened dipolar interaction, which eventually increases the reaction rate. To confirm the importance of ordered arrangement of modified cations, we measured $R_{ct}$ of CP modified by quaternary ammonium cations with different chain lengths (BTAB < OTAB < DTAB < CTAB) or different head sizes (TMAB < TBAB and HDBAB < EHDAB), considering that faster electron transfer is reflected by a lower $R_{ct}$ value. As a result, the quaternary ammonium cations with longer chains or smaller head sizes, thus with higher order of arrangement, exhibit lower $R_{ct}$ values (Fig. 4i), demonstrating our assumption. Since the PECT step is the RDS of electrochemical benzaldehyde C–C coupling under our tested conditions, we expect that the reaction rate is positively related to $1/R_{ct}$. When correlating these

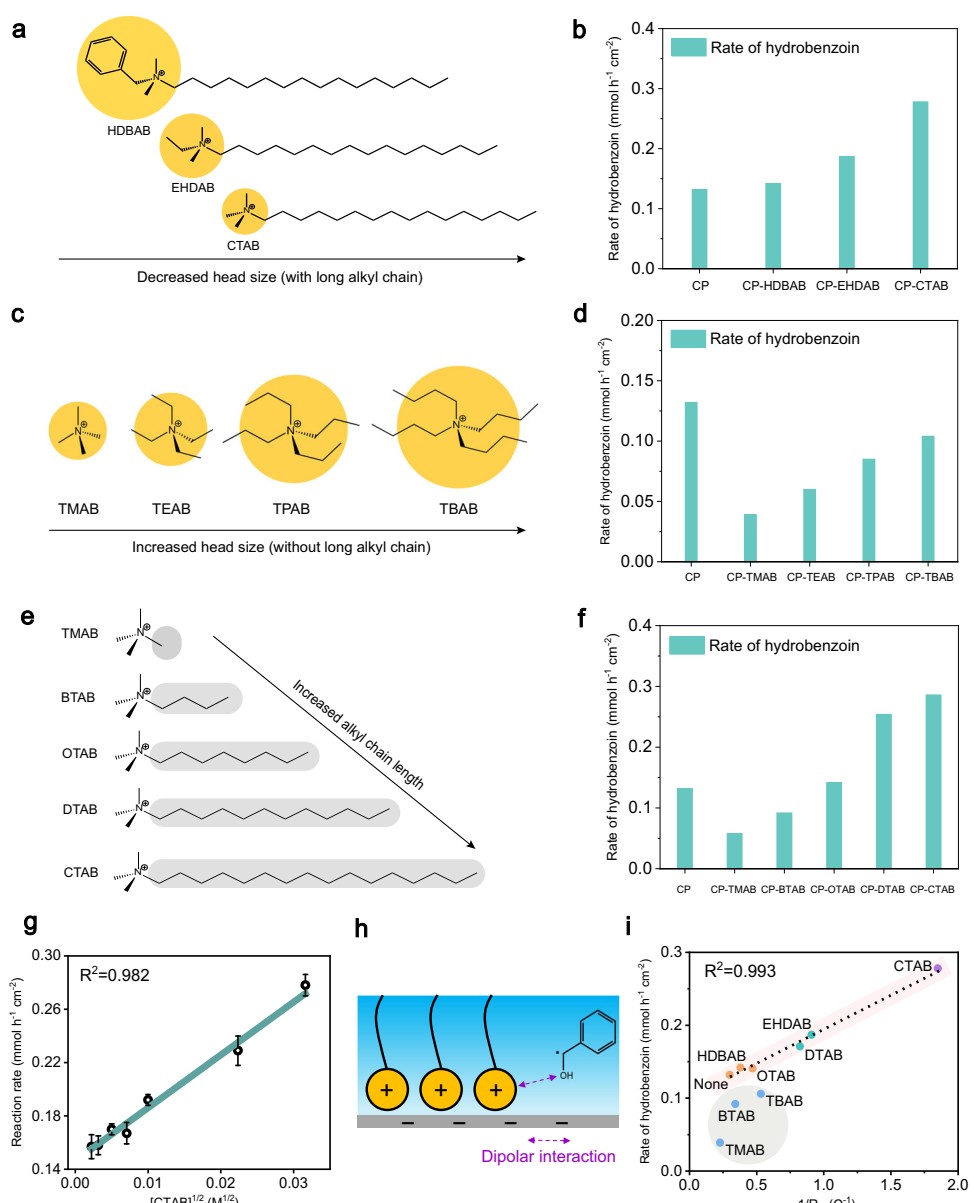

**Fig. 4 | Understanding of dipolar interaction between the quaternary ammonium cations and reaction intermediates.** CP modified with **a** quaternary ammonium cations with different head sizes and the same long chain, and **b** corresponding reaction rate of hydrobenzoin. CP modified with **c** quaternary ammonium cations with different head sizes but without long chains, and **d** corresponding reaction rate of hydrobenzoin. CP modified with **e** quaternary ammonium cations with the same head size but different chain lengths, and **f** corresponding reaction rate of hydrobenzoin. All the reactions were conducted at −1.4 V versus Ag/AgCl in 0.5 M Na$_2$SO$_4$ electrolyte. **g** Correlation plot between the concentration of CTAB and reaction rate. Error bars correspond to the standard deviation of three independent measurements. **h** Schematic of dipolar interaction between the head group of CTAB and ketyl radical. **i** Relationship between 1/R$_{ct}$ and reaction rate. A good linearity is obtained for quaternary ammonium cations with long chains (light pink region), while a negative deviation from linearity is observed for quaternary ammonium cations with short chain (light gray region).

two parameters, we obtain a good linearity for CP modified with quaternary ammonium cations with long chains (Fig. 4i). However, for CP modified with short-chain cations, a negative deviation from the linear relationship is displayed. The degree of deviation is negatively correlated with the size of cation head (Fig. 4c), which can be tentatively attributed to the blockage of the CP interface by the quaternary ammonium cations via site-blocking effect[39] (Supplementary Note 4).

Furthermore, to unveil the underlying reason for the linear correlation between 1/R$_{ct}$ and reaction rate, we attempted to explain the physical implication of 1/R$_{ct}$ according to transition state theory, namely Marcus theory[40,41]. We deduce that R$_{ct}$ is influenced by applied potential and the thermal rate constant ($K$) without applied potential ($\varphi = 0$), in which $K$ is related to $E_a$ of benzaldehyde C–C coupling

reaction without external potential. We then measured $E_a$ with external potential ($\varphi = −1.4$ V versus Ag/AgCl), and it showed a significant decrease after CTAB was introduced (from 0.591 to 0.233 eV), verifying that introducing CTAB decreases R$_{ct}$ and gives lower $E_a$ value. This can be understood by the dipolar interaction between CTAB and ketyl radical that decreases the energy of the transient state, leading to a lower $E_a$ value (see Supplementary Figs. 28–30 and Supplementary Note 5 for more discussions).

**Understanding of enhanced racemate stereoselectivity**

After elucidating the promoted activity by CTAB, we diverted our attention to understanding the increased racemate stereoselectivity over CP-CTAB (Fig. 2b). According to previous mechanistic study[42],

electrochemical reduction of benzaldehyde via a PCET process gives corresponding ketyl radicals before further C−C coupling takes place. Thus, the stereoselectivity is more likely determined by the conformational relation of two ketyl radicals with prochiral centers.

Without CTAB, a water layer is formed in the IHP via direct adsorption of water over the CP surface (Fig. 5a, left). Owning to the strong solvation effect of abundant interfacial water molecules on the hydroxyl groups of the ketyl radicals, we deduce that intramolecular hydrogen bonding between hydroxyl groups of two ketyl radicals is largely inhibited, thereby mesomer is energetically more favorable (Fig. 5a, right, Newman projection is used to present the structure of hydrobenzoins). The decreased stereoselectivity of racemate at higher temperature further validates this possibility (Supplementary Fig. 31), because thermal motion becomes stronger at higher temperature alleviating hydrogen bonding strength. In contrast, in the presence of CTAB, interfacial water molecules are repelled away from the interface because of the hydrophobicity endowed by the orderly-arranged hexadecyl group in CTAB (Fig. 5b, left). As a consequence, the hydrogen bonding between the hydroxyl groups of two ketyl radicals becomes more prominent, facilitating racemate generation (Fig. 5b, right).

To demonstrate the influence of CTAB on regulating the hydrophobicity at electrode−electrolyte interface, in situ ATR-SEIRAS spectra related to the presence of interfacial water were analyzed (Fig. 5c). The band at 1687 cm⁻¹ (dash line) is assigned to the H−O−H bending mode and the broad band spanning from 3000 to 3800 cm⁻¹ (region highlighted in blue) is assigned to the O−H stretching mode[43]. Furthermore, the O−H stretching band can be deconvolved into three types of O−H stretching vibrations. Specifically, the components located at ~3250 cm⁻¹, ~3390 cm⁻¹ and ~3620 cm⁻¹ are assigned to hydrogen-bonded water with saturated coordination, hydrogen-bonded water with unsaturated coordination, and isolated water, respectively. As the potential goes negative, the broad band of ν-OH displays an obvious blue shift, indicating the increase of low-coordination water (hydrogen-bonded water with saturated coordination and isolated water), suggesting the formation of hydrophobic microenvironment, consistent with the ordered arrangement of CTAB molecules at more negative potential as demonstrated above.

To verify the importance of interface hydrophobicity to repel interfacial water, acid-untreated CP (thus, electrode−electrolyte interface is hydrophobic) was then employed as the electrode. To our expectation, the stereoselectivity of racemates was also improved

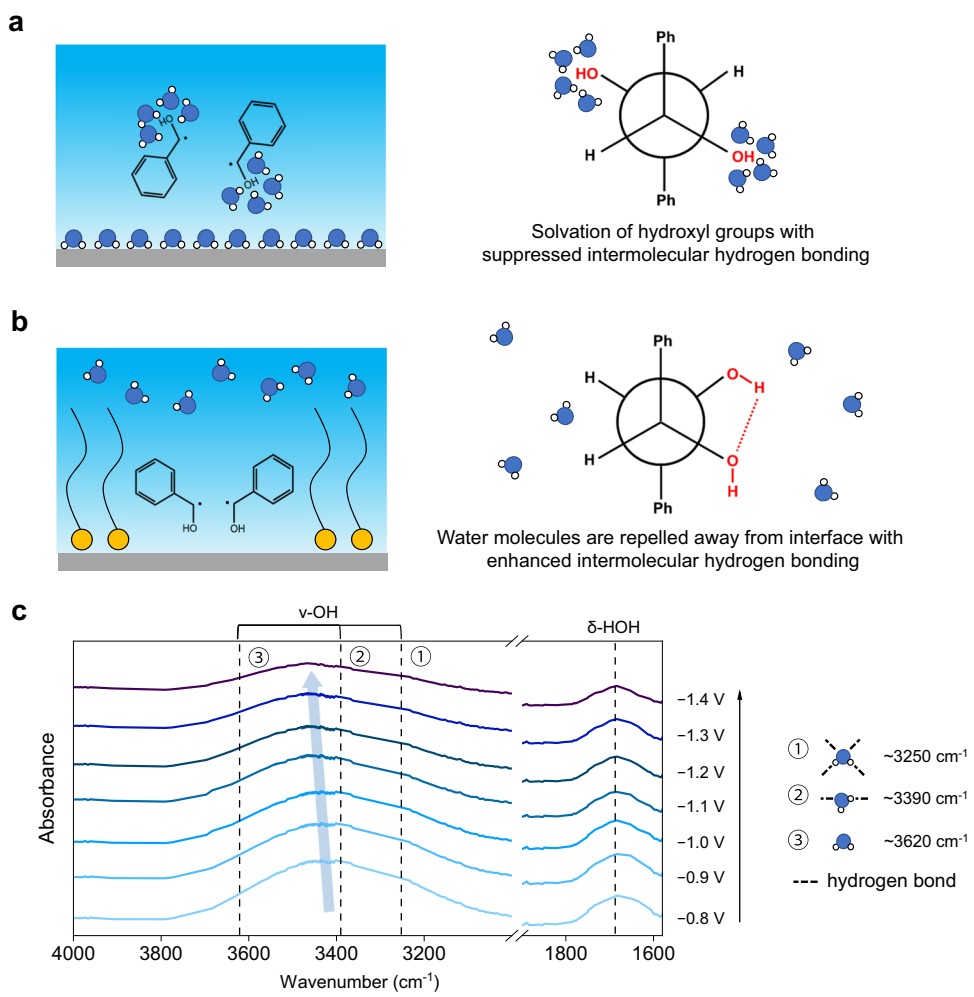

**Fig. 5 | Understanding of the enhanced racemate stereoselectivity by CTAB modification.** Schematic of electrode−electrolyte interface and corresponding C−C coupling products: **a** Without CTAB. A water layer is formed at electrode−electrolyte interface. The interfacial water molecules show a strong solvation effect on the hydroxyl group of the ketyl radicals, which favors mesomer generation. **b** With CTAB. Interfacial water molecules are repelled away from electrode−electrolyte interface, which favors racemates generation. **c** In situ ATR-SEIRAS spectra at different potentials in the range of 1600–1800 cm⁻¹ and 3000–4000 cm⁻¹ over CP-CTAB electrode, in which the dash lines show the signals of interfacial water, and the light-blue arrow guides the blue shift in wavenumber at more negative potentials. Three representative interfacial water types are shown on the right, including hydrogen-bonded water with saturated coordination, hydrogen-bonded water with unsaturated coordination, and isolated water.

compared with that over acid-treated CP (electrode–electrolyte interface is hydrophilic, the one used in the above studies), and their reaction rates are very close to each other (Supplementary Fig. 27). Furthermore, substrates with increased hydrophobicity (furfural < benzaldehyde < acetophenone, Fig. 6a, b), surfactants with smaller head size and thus less steric repulsion (HDBAB < EHDAB < CTAB, Fig. 6c) or surfactants with longer chain length and thus stronger hydrophobic interaction (TMAB < BTAB < OTAB < DTAB < CTAB, Fig. 6d) all deliver higher racemate stereoselectivity, unambiguously demonstrating the importance of hydrophobic interface to repel interfacial water, suppress solvation of ketyl radical and facilitate hydrogen bond between two ketyl radicals, which in turn favors racemate production (Fig. 6e and Supplementary Fig. 32). To consolidate this rationale, we prepared mixed solvents ($H_2O$–$C_2H_5OH$, $H_2O$–$CH_3CN$ with volume/volume ratio of 3:1) for electrolysis. Note that the solvation extent for hydroxyl groups follows a sequence of $CH_3CN$ < $C_2H_5OH$ < $H_2O$, depending on the corresponding strength of hydrogen bond and dipolar interaction[44]. We anticipate that the solvation effect on the ketyl radical would be mitigated by using the mixing solvents compared with that using pure water. As expected, the stereoselectivity of racemate increases when the mixed solvents were used (Supplementary Fig. 33) or when the ratio of low-solvation solvent ($CH_3CN$) to water increased (Supplementary Figs. 34 and 35).

We recognize mesomer stereoselectivity was not fully suppressed by CTAB modification, which is not unreasonable because controlling the conformation of transient ketyl radicals remains challenging. Meanwhile, the relatively weak binding energy of hydrogen bond (typically ~20 kJ/mol)[45] suggests that either the interaction between hydroxyl groups (that favors racemate) or between the hydroxyl group and water (that favors mesomer) may not be strong enough to prevail. More studies can be carried out in the future to design a micro-environment with stronger interaction and spatially more optimized configuration with the reactant and/or intermediates, which we anticipate would be efficient to further enhance the stereoselectivity.

## Discussion

In conclusion, we reported an efficient microenvironment regulation strategy by modification of CP using CTAB, creating an optimal microenvironment at electrode–electrolyte interface, delivering a 2.2-fold higher reaction rate and unprecedented 2.5-fold higher racemate stereoselectivity in electrochemical pinacol C–C coupling reaction. Based on the collective evidence, the plausible reaction mechanism is proposed (Fig. 7). Part I: Without CTAB, benzaldehyde molecules orientate randomly and water layer is formed over the CP interface. Part II: In the presence of CTAB, CTAB is adsorbed over the CP interface using its head group under negative potential, leaving its long-chain

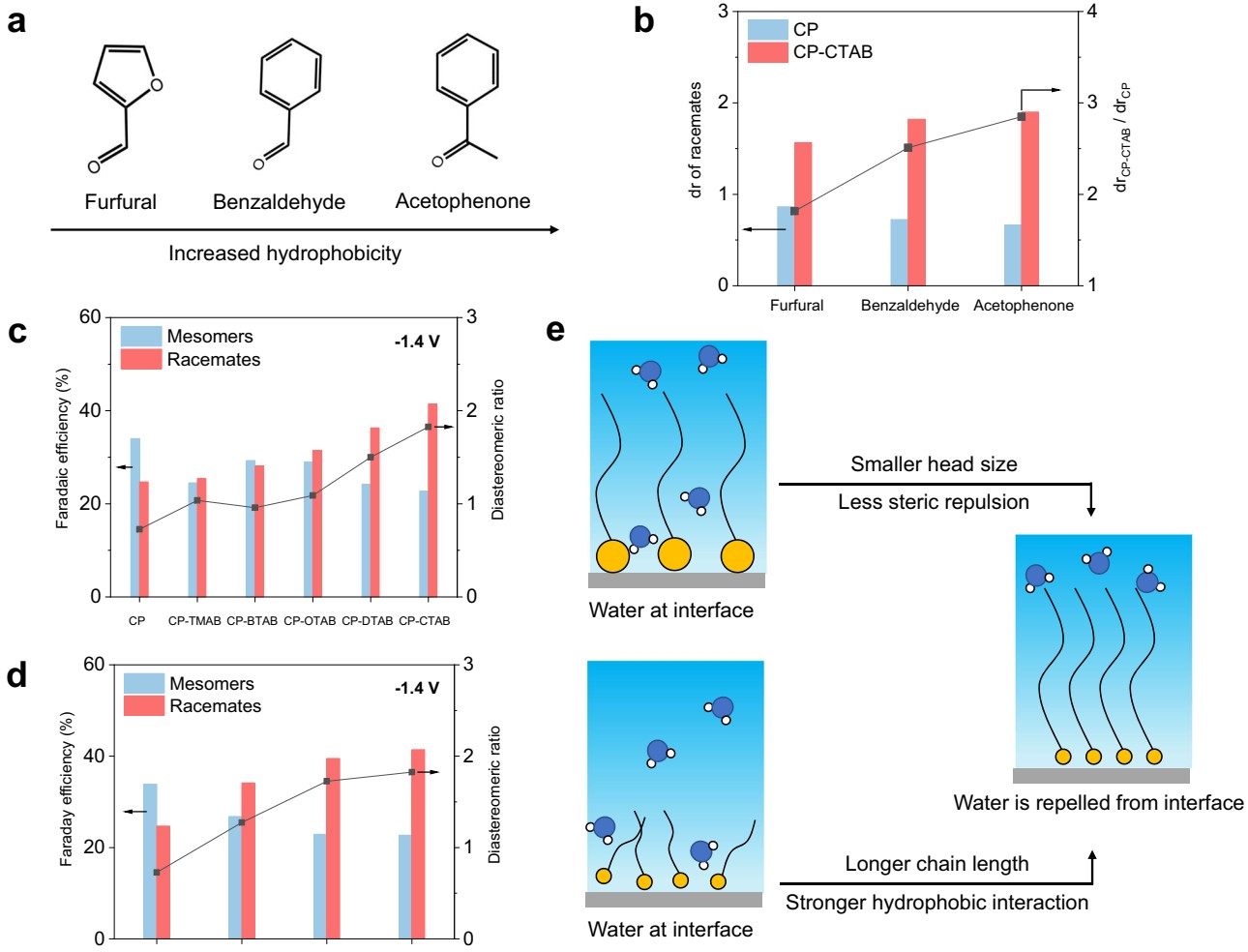

**Fig. 6 | Effects of different reactants and quaternary ammonium cations on racemates stereoselectivity. a** Structures of different substrates with increased hydrophobicity, **b** corresponding stereoselectivity results. Stereoselectivity of hydrobenzoin over CP modified with quaternary ammonium cations containing **c** different head groups and **d** different tail chains at −1.4 V versus Ag/AgCl in 0.5 M $Na_2SO_4$ electrolyte. **e** Schematic illustration of the effects of different head groups and tail chains of quaternary ammonium cations on repelling interfacial water away from electrode.

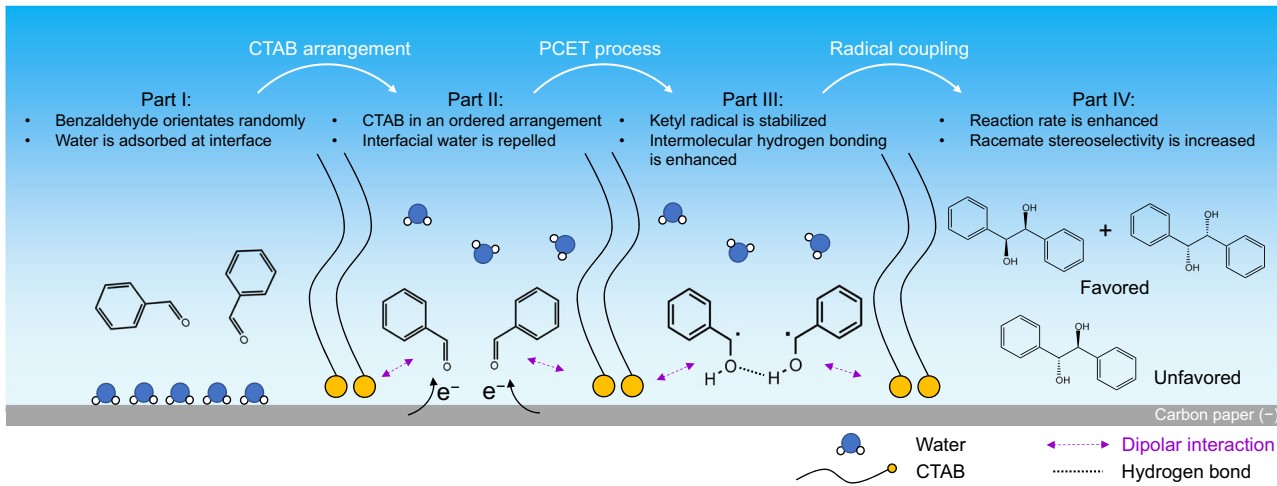

**Fig. 7 | Proposed reaction mechanism of electrochemical benzaldehyde C–C coupling reaction.** The reaction process is depicted in four parts (from left to right). Part I: without CTAB, reactant orientates randomly and water layer is formed over CP surface. Part II: CTAB is adsorbed over CP interface in an ordered arrangement, which enhances dipolar interaction with reactant and repels interfacial water. Part III: at more negative potential, ketyl radical is generated via a PCET process, and it is stabilized by CTAB at CP surface via dipolar interaction. Moreover, the conformation of ketyl radical is modulated by repelling away interfacial water molecules, with enhanced intramolecular hydrogen bonding. Part IV: Radical coupling takes place with enhanced activity and racemate stereoselectivity.

facing to electrolyte with an ordered arrangement. The formed microenvironment enhances dipolar interaction between the head group of CTAB and benzaldehyde and also repels interfacial water. Part III: At more negative potential (under electrochemical reaction condition), ketyl radical is generated as the reaction intermediate via a PCET process, and it is stabilized by CTAB via dipolar interaction. Meanwhile, the hydrogen bonds between two ketyl radicals are more prominent due to fewer interfacial water molecules at electrode–electrolyte interface. Part IV: As a result of dipolar interaction and interfacial water-repelling, the presence of CTAB enhances the reaction rate, and producing racemates is more favorable as the C–C radical coupling products.

Recently, alkali cation was reported to greatly modulate the microenvironment at electrode–electrolyte interface, affecting the catalytic performance[46]. Hence, electrocatalytic pinacol coupling was further conducted in the presence of different alkali cations with or without CTAB, aiming to examine if CTAB exhibits a unique effect on the catalytic performance. When CTAB was absent, different cations indeed affected the activity but to a less significant extent compared with CTAB (Supplementary Fig. 36). Meanwhile, the addition of $Cs^+$ promoted dr value to a higher extent compared with the addition of CTAB, while other cations did not exhibit a significant effect. The effect of $Cs^+$ on catalytic performance is worth further exploration, which is beyond the scope of this work. When CTAB was introduced, the activity and dr of hydrobenzoin were basically unchanged in the presence of different alkali cations, indicating that the specific adsorption of $CTA^+$ was more favorable under our reaction conditions. Collectively, these results suggest that CTAB serves as a unique molecule for microenvironment regulation in our study (see Supplementary Note 6 for more discussion).

In addition, it should be admitted that no preferential enantioselectivity was obtained by CTAB modification (RR:SS ratio of 1:1), because enantioselectivity could only be regulated by external chiral environment, as exemplified by the development of chiral intermediate-mediated strategies toward asymmetric radical reactions, including singly occupied molecular orbital (SOMO) catalysis[47], hydrogen bond catalysis[48] and metal complex catalysis[49]. For heterogeneous systems, recent work showed that stereoselectivity can be modulated by interfacial modification of chiral molecules[50,51]. Therefore, we envision that future efforts to create a chiral

microenvironment by employing chiral surfactants will hold great promise to enable enantioselectivity control.

## Methods

### Pretreatment of carbon paper (CP)
CP was cut into pieces of 2.5 × 1.0 cm. Then, CP was ultrasonicated with acetone for 5 min and subsequent water several times to remove organic impurities over the surface. Then, CP was immersed into the mixed solution consisting of DI water (5 mL), $H_2SO_4$ (5 mL, 98 wt%) and $HNO_3$ (5 mL, 68 wt%). The mixture was heated to 60 °C and maintained at this temperature for 24 h to improve its hydrophilicity.

### Pretreatment of Carbon ECP600JD
The Carbon ECP600JD powder (20 mg) was immersed into the mixed solution consisting of DI water (10 mL), $H_2SO_4$ (10 mL, 98 wt%) and $HNO_3$ (10 mL, 68 wt%). The mixture was heated to 80 °C with stirring and maintained at this temperature for 24 h to improve its hydrophilicity.

### Chemicals
Cetyl trimethyl ammonium bromide (CTAB, 99%), Dodecyl trimethyl ammonium bromide (DTAB, 99%), Octyl trimethyl ammonium bromide (OTAB, 99%), Butyl trimethyl ammonium bromide (BTAB, 99%), Tetramethyl ammonium bromide (TMAB, 99%), Tetra ethyl ammonium bromide (TEAB, 99%), Tetra-n-propyl ammonium bromide (TPAB, 99%), Tetra-n-butyl ammonium bromide (TBAB, 99%), Hexadecyl dimethyl benzyl ammonium bromide (HDBAB, 99%), Ethyl hexadecyl dimethyl ammonium bromide (EHDAB, 99%), $NaAuCl_4\cdot2H_2O$ (99.8%), $NH_4Cl$ (99%), $Na_2SO_3$ (98%), $Na_2S_2O_3\cdot5H_2O$ (98%) and NaOH (98%) were purchased from Macklin. $Na_2SO_4$ (99%), $Li_2SO_4$ (99%), $K_2SO_4$ (99%), $Cs_2SO_4$ (99%), Sodium dodecyl sulfate (SDS, 99.9%), NaBr (99%), Benzaldehyde (99%) and Hydrobenzoin (99%, RR, SS and meso isomers) were purchased from Aladdin. Carbon ECP600JD was purchased from Sinero.

### Characterizations
The XRD patterns were collected on a Bruker D8 ADVANCE with Cu Kα radiation (λ = 1.5406 Å) at 40 kV and 40 mA. The XPS analysis was measured on a Thermo EXCALAB 250Xi apparatus with a monochromatic Al Kα X-ray source. All data was calibrated by the C 1s peak at

284.8 eV. The electron paramagnetic resonance (EPR) spectra signals of radicals were detected on a JEOL FA-200. The contact angle experiments were carried out on OCA15Pro. The ATR-SERIRS measurements were achieved onto Bruker FT-IR spectrometer equipped with a liquid nitrogen-cooled MCT detector and a Si attenuated total reflection accessory. The spectral resolution was set to 2 cm$^{-1}$. The data was collected with 40 s resolution per spectrum and measured simultaneously by chronoamperometry technique between −0.8 V to −1.4 V. Reference spectra for the SEIRAS measurements were recorded at OCP in 0.5 M Na$_2$SO$_4$ electrolyte with 25 mM benzaldehyde and 1 mM CTAB. Au nanofilm was obtained by chemical deposition method. To strengthen the surface-enhancement effect of infrared spectroscopy, it is necessary to increase the contact area between carbon catalysts and Au nanofilm. Therefore, we chose Ketjen black powder (Carbon ECP600JD, abbreviated as CE) as the reaction electrode for ATR-SERIRS experiments, instead of the self-supporting CP electrode with less catalyst-nanofilm interface (Supplementary Fig. 37).

## Electrochemical measurements

The electrochemical performances of all the samples were tested using a CHI 760E electrochemical analyzer (CH Instruments, Inc., Shanghai). All the measurements were carried out in an H-type cell (40 mL per chamber) with Nafion 117 membrane. The three-electrode system consists of an Ag/AgCl electrode, a Pt counter electrode, a working electrode with as-prepared catalysts. The foams of the working electrode were cut into the size of 1.0 cm × 2.5 cm, and the geometric surface area for electrocatalytic tests is 2.0 cm$^2$ (2.0 cm × 1.0 cm). The electrolyte comprised 0.5 M Na$_2$SO$_4$ and 1 mM of the different surfactants, dissolved in deionized water (pH = 7.2), respectively. All the tests for electrochemical reduction of benzaldehyde were conducted under magnetic stirring of 800 rpm. Electrodes after electrolysis were briefly rinsed with deionized water, and completely dried under vacuum before measurement.

## Normal pulse voltammetry (NPV) experiments

The current is recorded 50 s after each potential step over the range of potentials from −0.85 to −2.00 V versus Ag/AgCl with the step width of 0.05 V and the applied potentials last for 0.05 s. All the tests for electrochemical reduction of benzaldehyde were conducted under magnetic stirring of 800 rpm.

## Electrochemical impedance spectroscopy (EIS) measurements

Electrochemical impedance spectroscopy measurements were performed in an H-type cell and the presence and absence of 1 mM different surfactants with 25 mM benzaldehyde in 0.5 M Na$_2$SO$_4$ electrolytes. Potential variation experiments were carried out from −0.8 to −1.4 V versus Ag/AgCl each within the frequency range between 10$^4$ Hz and 1 Hz and 5 mV amplitude. Tests were performed at 0.1 V intervals, with each test interval being 180 s. For differential capacitance experiment, the experiments were performed at the frequency of 10$^3$ Hz from −0.8 to −1.4 V versus Ag/AgCl. Temperature variation experiments were performed from the range of 298 K to 318 K and the other conditions were the same with potential variation experiments. The $\varphi_{peak}$ was understood either by the equivalent Nyquist plot that associated smaller Rct or the mathematical derivation using Eq. (4)

$$\varphi_{peak} = \frac{2R_{ct}}{5\sqrt{R_s(R_s + R_{ct})}} \qquad (4)$$

Double layer capacitances were calculated using Eq. (5), where Cdl is the double layer capacitance, Rct is the charge transfer resistance, CPE is the constant phase element, $n$ is the fitting parameter which was

also found after the fitting of the Nyquist plots. The relevant parameter values were fitted by ZView 2 software.

$$C_{dl} = \frac{(CPE \times R_{ct})^{\frac{1}{n}}}{R_{ct}} \qquad (5)$$

## Product quantifications

The sample was collected and diluted with water and then analyzed by HPLC (Agilent 1200 Infinity Series) equipped with a variable wavelength detector (VWD) at 210 nm. Benzaldehyde, furfural, acetophenone and their products were quantitatively analyzed using a C18 column (4.6 mm × 250 mm, 5 μm). For benzaldehyde and acetophenone, the column was operated at 35 °C with a flow rate of 1.0 mL min$^{-1}$ using CH$_3$CN-H$_2$O mixture (40%:60%, v/v) as the mobile phase. For furfural, the column was operated at 45 °C with a flow rate of 0.8 mL min$^{-1}$ using a binary gradient pumping method. The binary gradient pumping method was set as: the CH$_3$CN fraction in CH$_3$CN-water mixture (v/v) was kept at 15% (0–3.78 min), increased from 15 to 60% (3.78–11.28 min), kept at 60% (11.28–12.78 min), decreased from 60 to 15% (12.78–15 min), and kept at 15% (15–18 min). The isomers of hydrobenzoin were purchased and identified. The dimer of furfural and acetophenone was quantified by setting the response coefficient of the dimer to twice that for the corresponding alcohol, which was adopted by Xu's report[52]. Due to the commercial unavailability of the coupling products of furfural and acetophenone, $^1$H-NMR experiments were carried out to qualitatively identify stereoisomers, see Supplementary Figs. 3–6. The Faradic efficiency (FE) was calculated with the following equations:

$$FE(\%) = 100\% \times \frac{mole\ of\ product}{total\ charge\ passed / (n \times 96485C\ mol^{-1})} \qquad (6)$$

Where $n$ is the number of transferred electrons for each product, in which $n = 2$ for benzyl alcohol and each isomer of hydrobenzoin. The reaction rate of hydrobenzoin was calculated as follows:

$$Reaction\ Rate\left(mmol\ cm^{-2}\ h^{-1}\right) = \frac{mmol\ of\ total\ hydrobenzoin}{area \times reaction\ time} \qquad (7)$$

Where the area was the geometric area of electrodes (2.0 cm$^2$) and the reaction time is 1 h.

## Chemical deposition method for Au nanofilm

To conduct in situ attenuated total reflection surface-enhanced infrared absorption spectroscopy (ATR-SEIRAS) experiments, a polycrystalline Au nanofilm was deposited chemically onto a Si ATR-IR prism. The Si prism was first polished using a 0.05 μm alumina solution until the reflecting surface became hydrophobic. The alumina powder was then washed away from the crystal surface by alternative sonication in water, acetone, and then water. The treated crystal surface was then immersed in a 40% NH$_4$F solution for 5 min to remove the virgin oxide layer and to create a hydride-terminated surface. The reflecting surface was then immersed in a plating solution at 55 °C for 5 min to deposit the Au film. The plating solution is a mixture of 3 mL 5 wt% HF and a gold plating solution containing NaAuCl$_4$·2H$_2$O (0.009 M), NH$_4$Cl (0.03 M), Na$_2$SO$_3$ (0.09 M), Na$_2$S$_2$O$_3$·5H$_2$O (0.03 M) and NaOH (0.03 M).

# Data availability

All data that support the findings of this study are available in the paper and its Supplementary Information or from the corresponding author upon request. Source data are provided with this paper.

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

## Acknowledgements

This work was supported by the Beijing Natural Science Foundation (JQ22003), the National Natural Science Foundation of China (Grant No. 22325805, 21978147, 21935001), and the Haihe Laboratory of Sustainable Chemical Transformations.

## Author contributions

K.K. and H.D. conceived the project and co-wrote the manuscript. K.K. carried out most of the experiments. A.L. helped with sample characterizations, contributed to the ATR-SEIRAS and EPR experiments, and participated in the early discussion on stereoselectivity and kinetic analysis. Y.W. contributed to the deposition of Au nanofilms, ATR-SEIRAS experiments and revision of Supplementary Information. Q.S. contributed to EPR experiments and provided useful suggestions for figure editing. K.J. contributed to the contact angle experiment. K.J. and J.L. provided useful suggestions on experiment designs.

## Competing interests
