## [Peer Review File · Nature Communications]

Electrochemical carbon–carbon coupling with enhanced activity and racemate stereoselectivity by microenvironment regulationReviewers' Comments:

Reviewer #1:

Remarks to the Author:

This manuscript describes a microenvironment regulation strategy which not only enhanced the reaction rate of electroreductive C-C coupling of benzaldehyde but also regulated the stereoselectivity of hydrobenzoin products. The authors showed the establishment of a confined microenvironment consisting of ordered CTAB molecules at electrode/electrolyte interface under reaction conditions. Based on this knowledge and substantial evidences, the authors convincingly demonstrated that the enhanced dipolar interactions contributed to the promoted activity, and the hydrophobic microenvironment contributed to the regulated stereoselectivity. Overall, this work provides a very systematic understanding of the structure-activity relationship for the microenvironment regulation in electrocatalysis, and a novel and fascinating perspective on the stereoselectivity regulation of C-C coupling products. In my opinion, this work is insightful for the rational design of microenvironment regulation strategy beyond biomass electroreduction. Therefore, I recommend the publication of this manuscript in *Nat. Commun.* upon minor revisions.

1. In the introduction section, the statement: "this strategy was limited to a handful electrocatalytic reactions including hydrogen, oxygen and CO₂ reductions." requires additional references (*Chem* 2022, 8, 1-15 for electrocatalytic oxygen evolution reaction; *Angew Chem Int Ed* 2022, 61, e202113362 for glycerol electrooxidation) about the microenvironment strategy for electrooxidation reaction to demonstrate the universality of this strategy (*National Sci. Rev.* 2023, nwad149; 2021, 8, nwaa293).
2. A related work on stereoselectivity regulation in electrocatalytic aromatic aldehyde C-C coupling was mentioned by a literature of Kashimura et al. (*Electrochimica Acta* 2005, 51, 14-22). Although it did not provide any opinion on the structure-activity relationship of stereoselectivity regulation, I think this work should be cited in the introduction section.
3. The cation strategy was universally reported in electrocatalytic CO₂ reduction reaction (CO₂RR) for the different microenvironment created by local field of hydrated cations. Xu et al. proposed the CO₂RR activity was not related to the concentration of OH⁻, but rather to that of cations (*Angew. Chem. Int. Ed.* 2020, 59, 4464-4469). Thoi et al. also discovered that introduced CTAB molecules had different effects on CO₂RR activity in the electrolyte containing different cations (*ACS Catal.* 2020, 10, 9907-9914). Hence, I am curious whether the supporting electrolytes with different cations will have similar effects on the reported system in this manuscript, and how these cations will interact with the introduced CTA⁺ cations.
4. As far as I know, the stereoisomers of electroreductive product of furfural and acetophenone are not available to be purchased. Hence, I wonder how you distinguish between these two isomers of the products of furfural and acetophenone.
5. Supplementary Fig. 24d is not consistent with the main text, in which the stereoselectivity of acid-treated CP with CTAB should be similar with that of nonacid-treated CP with CTAB as the manuscript mentioned.

Reviewer #2:

Remarks to the Author:

In this work, Kong et al. described a microenvironment regulation strategy by modifying carbon paper with CTAB, delivering electrochemical C-C coupling of benzaldehyde with enhanced activity and racemate stereoselectivity. The understanding of dipolar interaction between CTAB and intermediates by employing surfactants with different head size and alkyl chain length are well organized and convincing. The hydrogen bond effect on racemate stereoselectivity is also well supported by experimental data and reasonable analysis. To my best knowledge, stereoselectivity regulation remains largely unexplored in electrocatalysis field, and this work is timely and may inspire more relative study in electrocatalysis community. I recommend its publication in this journal after the

authors address the following technical issues :

1. For acid-treated and untreated carbon paper, the contact angle (CA) experiment should be provided to prove the changes in interfacial hydrophobicity.
2. Whether the results of CA experiments were measured at the same moment during the experimental process? It's well-known that the contact angle will change with the measurement time.
3. Tafel experiments for CP and CP-CTAB require i-R correction to obtain the intrinsic electrochemical kinetic features.
4. It's noticed that in the ATR-SERIRS measurements, Carbon ECP600JD was used instead of CP for the experiment. It is necessary to compare the difference in electrocatalytic performance of Carbon ECP600JD and Carbon ECP600JD-CTAB in the H cell and compare them with the electrocatalytic systems with CP and CP-CTAB.
5. The qualitative and quantification method of coupling products of furfural and acetophenone should be provided given that their diastereomers are not commercially available.
6. Some typo issues, such as in Supplementary Tables 1-8, the units of R_s and R_{ct} should be $\Omega \cdot \text{cm}^2$ rather than $\Omega \cdot \text{cm}^{-2}$.

Response letter

General response: We thank the reviewers for their valuable comments and appreciate the editor for giving us the opportunity to revise the manuscript; a series of additional experiments have been performed to enable us to fully address the comments raised by the reviewers. Please find the point-by-point response below, the revised Manuscript and Supplementary Information with the changes highlighted in yellow.

Response to Reviewer #1:

This manuscript describes a microenvironment regulation strategy which not only enhanced the reaction rate of electroreductive C-C coupling of benzaldehyde but also regulated the stereoselectivity of hydrobenzoin products. The authors showed the establishment of a confined microenvironment consisting of ordered CTAB molecules at electrode/electrolyte interface under reaction conditions. Based on this knowledge and substantial evidences, the authors convincingly demonstrated that the enhanced dipolar interactions contributed to the promoted activity, and the hydrophobic microenvironment contributed to the regulated stereoselectivity. Overall, this work provides a very systematic understanding of the structure-activity relationship for the microenvironment regulation in electrocatalysis, and a novel and fascinating perspective on the stereoselectivity regulation of C-C coupling products. In my opinion, this work is insightful for the rational design of microenvironment regulation strategy beyond biomass electroreduction. Therefore, I recommend the publication of this manuscript in Nat. Commun. upon minor revisions.

Comment 1: In the introduction section, the statement: “this strategy was limited to a handful electrocatalytic reactions including hydrogen, oxygen and CO₂ reductions.” requires additional references (*Chem* 2022, 8, 1-15 for electrocatalytic oxygen evolution reaction; *Angew Chem Int Ed* 2022, 61, e202113362 for glycerol electrooxidation) about the microenvironment strategy for electrooxidation

reaction to demonstrate the universality of this strategy (*National Sci. Rev.* 2023, nwad149; 2021, 8, nwa293).

Response: We thank the reviewer for providing the valuable references. 4 additional papers on the microenvironment strategy have been added in the introduction section.

To clearly present these results, we revised the Manuscript as follows:

“These successful examples show the potential of microenvironmental regulation to enhance catalytic activity. In addition, this strategy exhibits a wide range of applications,^{10, 11} including electrocatalytic reductions (e.g., hydrogen evolution reaction (HER)¹², oxygen reduction reaction (ORR)¹³ and carbon dioxide reduction reaction (CO₂RR)¹⁴) and oxidations (e.g., oxygen evolution reaction (OER)¹⁵ and glycerol electrooxidation (GOR)¹⁶).” (Please see Page 2 in the revised Manuscript)

Comment 2: A related work on stereoselectivity regulation in electrocatalytic aromatic aldehyde C-C coupling was mentioned by a literature of Kashimura et al. (*Electrochimica Acta* 2005, 51, 14–22). Although it did not provide any opinion on the structure-activity relationship of stereoselectivity regulation, I think this work should be cited in the introduction section.

Response: We thank the reviewer for providing the valuable paper, which has been added in the introduction section.

To clearly present these results, we revised the Manuscript as follows:

“However, stereoselectivity regulation of electrocatalytic aromatic aldehyde C–C coupling was rarely studied¹⁷, with lacking of in-depth understanding of the structure-activity correlation. Meanwhile, microenvironment regulation has never been explored for ...” (Please see Page 2 in the revised Manuscript)

Comment 3: The cation strategy was universally reported in electrocatalytic CO₂ reduction reaction (CO₂RR) for the different microenvironment created by local field of hydrated cations. Xu et al. proposed the CO₂RR activity was not related to the concentration of OH⁻, but rather to that of cations (*Angew. Chem. Int. Ed.* 2020, 59, 4464–4469). Thoi et al. also discovered that introduced CTAB molecules had

different effects on CO₂RR activity in the electrolyte containing different cations (ACS Catal. 2020, 10, 9907-9914). Hence, I am curious whether the supporting electrolytes with different cations will have similar effects on the reported system in this manuscript, and how these cations will interact with the introduced CTA⁺ cations.

Response: According to these insightful suggestions, the effects of different alkali cations on the catalytic reaction were performed. Li₂SO₄, K₂SO₄ and Cs₂SO₄ at the same concentration (0.5 M; due to solubility limitation, the actual concentration of Li₂SO₄ was about 0.25 M) were used, respectively, as substitutes for Na₂SO₄ electrolyte. Then, electrochemical pinacol coupling was conducted with or without CTAB. The catalytic results are shown in the revised Supplementary Fig. 37.

Without CTAB, the reaction rate followed the trend of Li⁺ < Na⁺ < K⁺ ≈ Cs⁺. The dr of hydrobenzoin followed the trend of Na⁺ ≈ K⁺ < Li⁺ < Cs⁺. These results indicate that different cations indeed affected the activity and dr, probably induced by adsorption at electrode interface. However, the increased extent of reaction rate was not as significant as that using CTAB (K⁺ and Cs⁺: 0.18 mmol h⁻¹ cm⁻²; CTAB: 0.28 mmol h⁻¹ cm⁻²), suggesting the important role of CTAB for the enhanced activity. Regarding dr value, the addition of Cs⁺ promoted dr (2.14) even higher than the addition of CTAB (1.82), while other cations did not exhibit a significant effect. These results can be explained by the specific adsorption of Cs⁺ at electrode interface to repel the interfacial water, as evidenced by the desorption peak at -0.8 V versus Ag/AgCl in the LSV curve (the revised Supplementary Fig. 37a, b). We tentatively proposed that Cs⁺ may display a similar effect as CTA⁺, that is, creating a hydrophobic microenvironment at electrode–electrolyte interface.

When CTAB was introduced into the electrolyte involving Li⁺, Na⁺, K⁺ or Cs⁺ cations, the reaction rate exhibited an order of Li⁺ < Na⁺ ≈ K⁺ < Cs⁺ but with insignificant difference. The dr of hydrobenzoin was basically unchanged between different alkali cations. These results indicate that the specific adsorption of CTA⁺ was more favorable compared with that of other cations we tested.

Collectively, the above experiments indicate CTAB serves as a unique molecule

for microenvironment regulation during electrochemical pinacol coupling reaction. In addition, the varied dr induced by Cs^+ is worth further exploration, and its in-depth investigation is beyond the scope of this manuscript.

Revised Supplementary Fig. 36 Investigation of different alkali cations on affecting electrochemical pinacol C-C coupling reaction. a LSV plot and **b** corresponding enlarged region. Reaction conditions: electrolyte contains 0.5 M Cs_2SO_4

and 50 mM benzaldehyde, with or without CTAB. Reaction rate of hydrobenzoin in the electrolyte with 0.5 M different alkali cations (Li^+ , Na^+ , K^+ or Cs^+) at -1.4 V versus Ag/AgCl **c** without or **d** with 1 mM CTAB. Stereoselectivity of hydrobenzoin in the electrolyte with different cations at -1.4 V versus Ag/AgCl **e** without or **f** with 1 mM CTAB. Error bars correspond to the standard deviation of three independent measurements.

Based on the above discussion, we revised the Manuscript and Supplementary Information as follows:

1. The discussion on alkali cations on the catalytic performance has been added in the revised Manuscript:

“Recently, alkali cation was reported to greatly modulate the microenvironment at electrode–electrolyte interface, affecting the catalytic performance⁴⁶. Hence, electrocatalytic pinacol coupling was further conducted in the presence of different alkali cations with or without CTAB, aiming to examine if CTAB exhibits a unique effect on the catalytic performance. When CTAB was absent, different cations indeed affected the activity but with less significant extent compared with CTAB (Supplementary Fig. 36). Meanwhile, the addition of Cs^+ promoted dr value with higher extent compared with the addition of CTAB, while other cations did not exhibit a significant effect. The effect of Cs^+ on catalytic performance is worth further exploration, which is beyond the scope of this work. When CTAB was introduced, the activity and dr of hydrobenzoin were basically unchanged in the presence of different alkali cations, indicating that the specific adsorption of CTA^+ was more favorable under our reaction conditions. Collectively, these results suggest that CTAB serves as a unique molecule for microenvironment regulation in our study (see Supplementary Note 6 for more discussion.” (Please see Page 16 in the revised Manuscript)

2. The discussion of the effect of alkali cations on catalytic performance was presented in the revised Supplementary Note 6 in the Supplementary Information:

Li_2SO_4 , K_2SO_4 and Cs_2SO_4 at the same concentration (0.5 M; due to solubility

limitation, the actual concentration of Li₂SO₄ was about 0.25 M) were used, respectively, as substitutes for Na₂SO₄ electrolyte. Then, electrochemical pinacol coupling was conducted with or without CTAB. The catalytic results are shown in the revised Supplementary Fig. 36.

Without CTAB, the reaction rate followed the trend of Li⁺ < Na⁺ < K⁺ ≈ Cs⁺. The dr of hydrobenzoin followed the trend of Na⁺ ≈ K⁺ < Li⁺ < Cs⁺. These results indicate that different cations indeed affected the activity and dr, probably induced by adsorption at electrode interface. However, the increased extent of reaction rate was not as significant as that using CTAB (K⁺ and Cs⁺: 0.18 mmol h⁻¹ cm⁻²; CTAB: 0.28 mmol h⁻¹ cm⁻²), suggesting the important role of CTAB for the enhanced activity. Regarding dr value, the addition of Cs⁺ promoted dr (2.14) even higher than the addition of CTAB (1.82), while other cations did not exhibit a significant effect. These results can be explained by the specific adsorption of Cs⁺ at electrode interface to repel the interfacial water, as evidenced by the desorption peak at -0.8 V versus Ag/AgCl in the LSV curve (the revised Supplementary Fig. 36a, b). We tentatively proposed that Cs⁺ may display a similar effect as CTA⁺, that is, creating a hydrophobic microenvironment at electrode-electrolyte interface.

When CTAB was introduced into the electrolyte involving Li⁺, Na⁺, K⁺ or Cs⁺ cations, the reaction rate exhibited an order of Li⁺ < Na⁺ ≈ K⁺ < Cs⁺ but with insignificant difference. The dr of hydrobenzoin was basically unchanged between different alkali cations. These results indicate that the specific adsorption of CTA⁺ was more favorable compared with that of other cations we tested.

Collectively, the above experiments indicate that CTAB serves as a unique molecule for microenvironment regulation during electrochemical pinacol coupling reaction. In addition, the varied dr induced by Cs⁺ is worth further exploration, and its in-depth investigation is beyond the scope of this manuscript.” (Please see Page 7 in the revised Supplementary Information)

Comment 4: As far as I know, the stereoisomers of electroreductive product of

furfural and acetophenone are not available to be purchased. Hence, I wonder how you distinguish between these two isomers of the products of furfural and acetophenone.

Response: We appreciated the reviewer for providing this critical suggestion. The dimers of furfural and acetophenone are indeed not commercially available. According to the report of Kim and his coworker (*J. Org. Chem.* 1998, 63, 5235-5239), the stereoisomers of the reduction coupling products of aromatic aldehydes and ketones could be qualitatively and quantitatively identified by $^1\text{H-NMR}$. It was demonstrated that the chemical shift of directly-bonded hydrogen atoms (for aldehydes) or hydrogen atoms on methyl group (for ketones) connected to the chiral center on the mesomers was 0.05~0.10 ppm higher than that of racemate.

Our experiment results were consistent with the above observations. We carried out bulk electrolysis experiments with furfural or acetophenone as the substrates. Then, 0.9 mL of the reaction solution was abstracted with the addition of 0.1 mL of D_2O for $^1\text{H-NMR}$ analysis. The $^1\text{H-NMR}$ spectra were presented in revised Supplementary Figs. 3-4:

Revised Supplementary Fig. 3 $^1\text{H-NMR}$ spectrum of the reaction after bulk electrolysis of furfural. The reaction was conducted at -1.4 V vs. Ag/AgCl. The spectrum is consistent with that in the previous report²².

The chemical structure of furfural and 1,2-Di(2-furyl)-1,2-ethanediol are shown on the right:

Furfural: $^1\text{H-NMR}$ (10% D_2O) δ 6.69 (m, 1H), 7.50 (m, 1H), 7.85 (m, 1H), 9.43 (s, 1H).

1,2-Di(2-furyl)-1,2-ethanediol: $^1\text{H-NMR}$ (10% D_2O) δ 4.95 (s, 2H, racemate), 4.99 (s, 2H, mesomer), 6.23 (m, 2H), 6.31 (m, 2H), 7.37 (m, 2H).

Revised Supplementary Fig. 4 $^1\text{H-NMR}$ spectrum of the reaction system after the bulk electrolysis of acetophenone. The reaction was conducted at -1.6 V vs. Ag/AgCl. The chemical structure of acetophenone and 2,3-Diphenyl-2,3-butanediol are shown on the right:

Acetophenone: $^1\text{H-NMR}$ (10% D_2O) δ 2.61 (s, 3H), 7.51-7.95 (m, 5H).

2,3-Diphenyl-2,3-butanediol: $^1\text{H-NMR}$ (10% D_2O) δ 1.49 (s, 6H, racemate), 1.55 (s, 6H, mesomer), 7.26-7.64 (m, 10H).

Moreover, we measured the dr value of the coupling products from furfural and acetophenone by $^1\text{H-NMR}$ based on the qualitative method as shown above. The results show a good consistent with the results measured by HPLC shown in the original manuscript (revised Supplementary Fig. 5). Therefore, it is safe to qualitatively identify the stereoisomers of the coupling products of furfural and acetophenone by $^1\text{H-NMR}$, and quantitatively measured the reaction results by HPLC (revised Supplementary Fig. 6).

Revised Supplementary Fig. 6 HPLC spectra of the reaction products. HPLC spectra of the reaction products with **a** benzaldehyde, **b** furfural and **c** acetophenone. The reactions were carried out in 0.5 M Na_2SO_4 electrolyte containing 25 mM substrate.

Revised Supplementary Fig. 5 Comparison of dr value determined by HPLC and ¹H-NMR. Bulk electrolysis of furfural and acetophenone were performed, and the products were subjected to HPLC and ¹H-NMR analysis.

Based on the above discussion, we revised the Manuscript and Supplementary Information as follows:

1. The analysis details of ¹H-NMR have been provided in Methods of the revised Manuscript:

“... Benzaldehyde, furfural, acetophenone and their products were quantitatively analyzed using a C18 column (4.6 mm×250 mm, 5μm). For benzaldehyde and acetophenone, the column was operated at 35 °C with a flow rate of 1.0 mL min⁻¹ using CH₃CN-H₂O mixture (40%:60%, v/v) as the mobile phase. For furfural, the column was operated at 45 °C with a flow rate of 0.8 mL min⁻¹ using a binary gradient pumping method. The binary gradient pumping method was set as: the CH₃CN fraction in CH₃CN-water mixture (v/v) was kept at 15% (0~3.78 min), increased from 15% to 60% (3.78~11.28 min), kept at 60% (11.28~12.78 min), decreased from 60% to 15% (12.78~15 min), and kept at 15% (15~18 min). The isomers of hydrobenzoin were purchased and identified. The dimer of furfural and acetophenone were quantified by

setting the response coefficient of dimer to twice of that for the corresponding alcohol, which was adopted by Xu's report⁵². Due to the commercial unavailability of the coupling products of furfural and acetophenone, ¹H-NMR experiments were carried out to qualitatively identify stereoisomers, see Supplementary Figs 3-6. ..." (Please see Page 20 in the revised Manuscript)

2. We revised the caption of Supplementary Fig. 5 in Supplementary Information:

"According to the report of Kim and coworker²³, the stereoisomers of the reduction coupling products of aromatic aldehydes and ketones could be qualitatively and quantitatively identified by ¹H-NMR. It was demonstrated that the chemical shift of directly-bonded hydrogen atoms (for aldehydes) or hydrogen atoms on methyl group (for ketones) connected to the chiral center on the mesomers was 0.05~0.10 ppm higher than that of racemate.

Hence, we measured the dr value of coupling products of furfural and acetophenone by ¹H-NMR based on the qualitative method shown above. The results show a good consistent with the results measured by HPLC. Therefore, it is safe to qualitatively identify the stereoisomers of the coupling products from furfural and acetophenone by ¹H-NMR, and to quantitatively measured the reaction results by HPLC." (Please see Page 12 in the revised Supplementary Information)

Comment 5: Supplementary Fig. 24d is not consistent with the main text, in which the stereoselectivity of acid-treated CP with CTAB should be similar with that of nonacid-treated CP with CTAB as the manuscript mentioned.

Response: We thank the reviewer for point out the mistake in Supplementary Fig. 24d. We have corrected it in revised Supplementary Fig. 27d accordingly. (Please see Page 34 in the Supplementary Information)

Response to Reviewer #2:

In this work, Kong et al. described a microenvironment regulation strategy by modifying carbon paper with CTAB, delivering electrochemical C-C coupling of benzaldehyde with enhanced activity and racemate stereoselectivity. The understanding of dipolar interaction between CTAB and intermediates by employing surfactants with different head size and alkyl chain length are well organized and convincing. The hydrogen bond effect on racemate stereoselectivity is also well supported by experimental data and reasonable analysis. To my best knowledge, stereoselectivity regulation remains largely unexplored in electrocatalysis field, and this work is timely and may inspire more relative study in electrocatalysis community. I recommend its publication in this journal after the authors address the following technical issues:

Comment 1: For acid-treated and untreated carbon paper, the contact angle (CA) experiment should be provided to prove the changes in interfacial hydrophobicity.

Response: We appreciated the reviewer for providing this suggestion. We conducted additional CA experiments and added the contact angle of untreated carbon paper (138°) in revised Supplementary Fig. 16.

Revised Supplementary Fig. 16 Contact angle experiments. Contact angle experiments of untreated CP, acid-treated CP and acid-treated CP-CTAB electrodes (from top to bottom). The photographs were captured at the moment when the water drop contacted with the carbon paper.

Comment 2: Whether the results of CA experiments were measured at the same moment during the experimental process? It's well-known that the contact angle will change with the measurement time.

Response: This is a meticulous suggestion. We have mentioned the testing details in revised Supplementary Fig. 16.

We revised the caption of revised Supplementary Fig. 16:

“The photographs were captured at the moment when the water drop contacted with the carbon paper”. (Please see Page 23 in the revised Supplementary Information)

Comment 3: Tafel experiments for CP and CP-CTAB require i-R correction to obtain the intrinsic electrochemical kinetic features.

Response: We appreciated the reviewer for providing this critical suggestion. We have carried out i-R correction on Tafel plot with the solution resistance measured by EIS experiments. The i-R corrected Tafel slopes of CP and CP-CTAB are 87.1 and 91.3 mV/dec, respectively (revised Supplementary Fig. 24). They are larger than 60 mV/dec, revealing that the RDS of benzaldehyde C–C coupling involves the transfer of the first electron, in agreement with the conclusion in the original manuscript.

Revised Supplementary Fig. 24 Tafel plots with i-R correction. Tafel plots with i-R correction of CP and CP-CTAB systems at a scan rate of 50 mV/s.

Comment 4: It's noticed that in the ATR-SERIRS measurements, Carbon ECP600JD was used instead of CP for the experiment. It is necessary to compare the difference in electrocatalytic performance of Carbon ECP600JD and Carbon ECP600JD-CTAB in the H cell and compare them with the electrocatalytic systems with CP and CP-CTAB.

Response: We thank the reviewer for this suggestion. To strengthen the surface-enhancement effect of infrared spectroscopy, it is necessary to increase the contact area between carbon catalysts and Au nanofilm. Therefore, we chose Ketjen black powder (Carbon ECP600JD, abbreviated as CE) as the reaction electrode for ATR-SERIRS experiments, instead of the self-supporting CP electrode with less catalyst-nanofilm interface. We used a similar acid treatment to CE to improve its hydrophilicity, dispersed it into ink (containing 2 mL ethanol, 50 μ L Nafion 117 and 5 mg acid-treated CE), and then sprayed the ink onto carbon paper as the electrocatalyst. Afterwards, bulk electrolysis experiments were carried out at -1.4 V versus Ag/AgCl without and with CTAB (marked as CE and CE-CTAB). The results show that a promoted activity and varied dr value were observed over CE electrode, similar with that over CP electrode (Figs. 2a, b), demonstrating that the conclusion of ATR-SERIRS over CE electrode is valid to explain the catalytic results over CP electrode.

Revised Supplementary Fig. 37 Catalytic performance over Carbon ECP600JD (CE) without or with CTAB. **a** Reaction rate and **b** stereoselectivity of hydrobenzoin over CE and CE-CTAB electrodes at -1.4 V versus Ag/AgCl. In the ATR-SERIRS experiments, Ketjen black powder (Carbon ECP600JD, abbreviated as CE) was used as a substitute for carbon paper.

Accordingly, we revised the Methods section in the revised Manuscript:

“**Characterization.** ... To strengthen the surface-enhancement effect of infrared spectroscopy, it is necessary to increase the contact area between carbon catalysts and Au nanofilm. Therefore, we chose Ketjen black powder (Carbon ECP600JD, abbreviated as CE) as the reaction electrode for ATR-SERIRS experiments, instead of the self-supporting CP electrode with less catalyst-nanofilm interface.” (Please see Page 18-19 in the revised Manuscript)

“**Pretreatment of Carbon ECP600JD.** The Carbon ECP600JD powder (20 mg) was immersed into the solution consisting of DI water (10 mL), H_2SO_4 (10 mL, 98 wt%) and HNO_3 (10 mL, 68 wt%). The mixture was heated to 80 °C with stirring and maintained at this temperature for 24 h to improve hydrophilicity.” (Please see Page 18 in the revised Manuscript)

We also added the discussion in the caption of Supplementary Fig. 37:

“The results show that a promoted activity and varied dr value were observed over CE electrode, similar with that over CP electrode (Figs. 2a, b), demonstrating that the conclusion of ATR-SERIRS over CE electrode is valid to explain the catalytic results over CP electrode.” (Please see Page 44 in the revised Supplementary Information)

Comment 5: The qualitative and quantification method of coupling products of furfural and acetophenone should be provided given that their diastereomers are not commercially available.

Response: It's such an insightful suggestion. We have provided the qualitative and quantitative methods for the coupling products of furfural and acetophenone as Reviewer 1# suggested. Please see the response to Comment 4 from Reviewer 1#.

Comment 6: Some typo issues, such as in Supplementary Tables 1-8, the units of R_s and R_{ct} should be $\Omega \cdot \text{cm}^2$ rather than $\Omega \cdot \text{cm}^{-2}$.

Response: We thank the reviewer for pointing this typo. We have corrected it in the revised Supplementary Tables 1-8, and also carefully examined the manuscript.

Reviewers' Comments:

Reviewer #1:

Remarks to the Author:

The authors have fully addressed my comments and revised manuscript is recommended for publication.

Reviewer #2:

Remarks to the Author:

This work can be published in current form.

We thank the reviewers for their positive comments for recommending the acceptance of the manuscript.